# Differential Visual Outcomes in Neovascular AMD Based on Ellipsoid Zone Integrity and Fluid Presence: Insights from a Phase III Trial

**DOI:** 10.3390/diagnostics15141815

**Published:** 2025-07-18

**Authors:** Justis P. Ehlers, Sari Yordi, Hasan Cetin, Reem Amine, Karen Matar, Asmita Indurkar, Katherine E. Talcott, Peter K. Kaiser, Arshad M. Khanani, Joanne Hu, Sunil K. Srivastava

**Affiliations:** 1The Tony and Leona Campane Center for Excellence in Image-Guided Surgery and Advanced Imaging Research, Cleveland Clinic, Cleveland, OH 44195, USA; sfyordi@gmail.com (S.Y.); cetinh@ccf.org (H.C.); aminer@ccf.org (R.A.); matark@ccf.org (K.M.); indurka@ccf.org (A.I.); talcotk@ccf.org (K.E.T.); srivass2@ccf.org (S.K.S.); 2Cole Eye Institute, Cleveland Clinic, Cleveland, OH 44195, USA; pkkaiser@gmail.com; 3Sierra Eye Associates, Reno, NV 89502, USA; arshad.khanani@gmail.com; 4School of Medicine, University of Nevada, Reno, Reno, NV 89557, USA; 5Novartis Pharmaceuticals Corporation, East Hanover, NJ 07936, USA; drjoannehu@gmail.com

**Keywords:** age-related macular degeneration, photoreceptor, retinal outer segment, subretinal fluid, intraretinal fluid

## Abstract

**Background/Objectives**: To investigate the effect of ellipsoid zone (EZ) integrity and retinal fluid on best-corrected visual acuity (BCVA) in neovascular, age-related macular degeneration. **Methods**: This was a post hoc treatment-agnostic analysis of the phase 3 HAWK trial. Intraretinal fluid (IRF), subretinal fluid (SRF), and ellipsoid zone (EZ) integrity were quantified over 48 weeks. EZ integrity maintenance was defined as EZ-RPE central subfield thickness (CST) >20 µm; partial EZ attenuation was EZ-RPE CST >0 and ≤20 µm; total EZ attenuation was EZ-RPE CST = 0 µm. **Results**: During treatment, BCVA in eyes with no fluid (66.5 to 70.2 letters) was greater than in eyes with IRF (59.5 to 62.4 letters) but comparable to BCVA in eyes with SRF (64.9 to 68.8 letters). In eyes with no fluid, BCVA was consistently greater in eyes with EZ integrity maintained (73.4 to 78.4 letters) than in eyes with EZ partial attenuation (65.3 to 66.5 letters) or with EZ total attenuation (55.8 to 59.8 letters). **Conclusions**: Eyes without fluid with EZ preservation achieved the highest overall BCVA, especially when compared to eyes without fluid and a lack of EZ preservation and to eyes with SRF. Achieving a “dry” status with preservation of EZ integrity is important in optimizing visual outcomes.

## 1. Introduction

Neovascular age-related macular degeneration (nAMD) is characterized by the development of fluid-leaking blood vessels in the subretinal pigment epithelium, subretinal or intraretinal space that is associated with vision loss [1,2,3]. If allowed to continue unabated, the invading vessels may ultimately become fibrosed lesions, and irreversible blindness may occur [1]. Treatment of nAMD with anti-vascular endothelial growth factor (anti-VEGF) therapies can reduce retinal fluid and improve visual acuity [3,4,5].

Several studies have examined the relationship between the presence of retinal fluid and visual acuity in eyes with nAMD. The presence of intraretinal fluid (IRF) has been associated with poorer visual acuity outcomes [5,6], whereas the presence of subretinal fluid (SRF) has been associated with similar or better visual acuity compared to the absence of SRF [5,6,7]. However, many of those studies mandated ongoing anti-VEGF therapy in the presence of SRF. Notably, recurrent SRF has been associated with a greater likelihood of visual acuity loss.

In addition to retinal fluid, imaging studies have revealed clear correlations between the integrity of retinal structures, such as the ellipsoid zone (EZ), and visual acuity [8]. In particular, eyes may have pre-existing atrophy that may not allow for visual recovery following fluid resolution. This creates a wide range of visual potential at treatment initiation that should be stratified to understand what specific features may be associated with the best BCVA outcomes. Being able to evaluate not only the presence/absence of fluid but also the integrity of the EZ in eyes that achieve SRF resolution might provide critical insights into overall therapeutic goals for fluid resolution. Such assessments are now feasible through advances in optical coherence tomography (OCT) analysis techniques that allow quantitative, in-depth characterization of retinal fluid and retinal integrity.

To address these issues, we examined the association between best-corrected visual acuity (BCVA) and retinal fluid presence, along with a concurrent assessment of EZ integrity in eyes without fluid with nAMD from the phase 3 HAWK study. This assessment provides important insights into the overall impact of EZ integrity in eyes that achieve “dry” status on BCVA outcomes and provides a comparative assessment for BCVA in eyes that have persistent fluid.

## 2. Materials and Methods

### 2.1. HAWK Study Design

HAWK (NCT02307682) was a 2-year, randomized, double-masked, multicenter trial conducted in 408 sites in North, Central, and South America; Europe; Asia; Australia; and Japan [4]. Protocols were approved by an independent ethics committee/institutional review board. All participants provided written informed consent. The detailed methodology of this study has been published previously [4]. Eligible participants were aged ≥50 years and had active, untreated, choroidal neovascular (CNV) lesions secondary to AMD that affected the central subfield in the study eye; CNV that comprised >50% of the total lesion area, assessed by fluorescein angiography; presence of IRF and/or SRF affecting the central subfield, assessed by spectral domain OCT (SD-OCT); and BCVA between 78 and 23 letters (inclusive) in the study eye, assessed by Early Treatment Diabetic Retinopathy Study (ETDRS) testing [4].

Study participants were randomized 1:1:1 to one of three treatment arms: brolucizumab 3 mg, brolucizumab 6 mg, or aflibercept 2 mg [4]. Brolucizumab was injected into the study eye at weeks 0, 4, and 8, which comprised the loading phase, with subsequent injections every 12 weeks unless disease activity was detected, in which case it was permanently adjusted to every 8 weeks. Aflibercept was injected every 8 weeks per label [4]. Participants received a full ophthalmic exam—including BCVA based on ETDRS testing and retinal fluid assessments—every 4 weeks, providing the data for the present analysis [4]. The primary endpoint was mean change in BCVA from baseline to week 48. Ultimately, the full analysis set included 358 participants for brolucizumab 3 mg, 360 for brolucizumab 6 mg, and 360 for aflibercept 2 mg.

### 2.2. Higher-Order Optical Coherence Tomography Analysis

SD-OCT scans (Cirrus, Zeiss [Oberkochen, Germany]; or Spectralis, Heidelberg Engineering [Heidelberg, Germany]) from the brolucizumab 6 mg and aflibercept 2 mg treatment groups collected during the HAWK study were sent to the Cleveland Clinic (Cleveland, OH, USA) for post hoc analysis. The HAWK trial included *n* = 360 eyes in the brolucizumab 6 mg arm and *n* = 360 eyes in the aflibercept 2 mg arm, of which 30 eyes and 38 eyes, respectively, were excluded for various reasons including image quality, availability, or compatibility. Scan acquisition for SD-OCT was the 6 × 6 mm macular cube with 128 B-scans for Cirrus and the 20° by 20° macular cube with 97 B-scans for Spectralis. Segmentation and identification of features on the SD-OCT scans utilized a previously described and validated process consisting of automated segmentation with a machine learning-based segmentation and feature extraction platform, followed by reviews and corrections or validation of segmentation by certified readers [9]. Previous validation showed inter-reader, intraclass correlation coefficients for metrics assessed with this process, including thickness from the EZ to the retinal pigment epithelium (EZ-RPE thickness), to be >0.9 in nAMD [10]. Each macular scan was automatically segmented to identify fluid boundaries and retinal layers of interest (i.e., EZ; Figure 1). A higher-order classifier then determined whether retinal fluid was SRF or IRF. Two masked certified readers evaluated segmentation accuracy, manually correcting any segmentation errors, as needed. The expert certified readers received the same SD-OCT analysis training, and the same reader reviewed and corrected any errors for a given participant across all time points, thus minimizing inter-time point and inter-reader variability. The reading environment was standardized with respect to location, computer configuration, monitor settings, and in-room lighting. The second senior image analyst then reviewed all scans to confirm accuracy and consistency of segmentation. Metrics exported for analysis at each time point included IRF and SRF volume across the macular cube and EZ-RPE central subfield thickness (CST). The central subfield was defined as the 1 mm diameter area centered on the fovea. EZ-RPE CST was calculated as the mean thickness of retinal tissue from the EZ to the RPE in the central subfield.

For the purposes of this post hoc analysis, EZ integrity maintenance was defined as EZ-RPE CST >20 µm. Partial EZ attenuation was defined as 0 µm < EZ-RPE CST ≤20 µm, and total EZ attenuation was defined as EZ-RPE CST = 0 µm (Figure 2). Eyes classified as having IRF could also exhibit SRF and vice versa; they were not mutually exclusive categories. Eyes that “achieved and maintained no fluid” were defined as eyes that achieved total reduction (i.e., 0 mm^3^) of IRF and SRF by week 16 and maintained total resolution of IRF and SRF by week 48.

### 2.3. Statistical Analyses

Differences in BCVA were analyzed using an analysis of covariance model. Depending on the analysis, EZ integrity group, fluid status, and treatment were used as factors and week 48 as a covariate. *p* < 0.05 was considered statistically significant. Statistical analyses were conducted using SAS 9.4 (SAS Institute Inc., Cary, NC, USA). Analyses were post hoc, and adjustments for multiple comparisons were not made because the analysis was intended to be hypothesis generating.

## 3. Results

### 3.1. Baseline Characteristics

A total of 652 eyes (one eye per participant) were included in this analysis (brolucizumab 6 mg: *n* = 330; aflibercept 2 mg: *n* = 322). Mean BCVA at baseline was 58.0 letters in eyes with IRF (SD = 13.8; *n* = 408) and 60.6 letters (SD = 13.8; *n* = 571) in eyes with SRF. In eyes with both IRF and SRF, mean BCVA at baseline was 57.8 letters (SD = 14.2; *n* = 336).

### 3.2. Visit-by-Visit Assessment of BCVA Based on Fluid Status and Ellipsoid Zone Integrity

At each visit, BCVA in eyes with IRF, SRF, or no fluid (i.e., no IRF or SRF) across the macula was determined. With anti-VEGF treatment, mean BCVA ranged from 59.5 to 62.4 letters for weeks 4 to 48 in eyes with IRF, from 64.9 to 68.8 letters in eyes with SRF, and from 66.5 to 70.2 letters in eyes with no fluid (Figure 3). At week 48, mean BCVA was significantly greater in eyes with no fluid (*n* = 343) than in eyes with IRF (*n* = 142; 70.2 vs. 60.5 letters, respectively; *p* < 0.0001), but BCVA in eyes without fluid showed no statistically significant difference from BCVA in eyes with SRF (*n* = 167; 68.1 letters; *p* = 0.22). Differences in mean BCVA for eyes with IRF, SRF, and no fluid maintained a similar pattern throughout treatment (Figure 3).

When evaluating EZ integrity within the central 1 mm subfield in eyes without fluid, mean BCVA ranged from 73.4 to 78.4 letters for weeks 4 to 48 in eyes with maintained EZ integrity (i.e., EZ-RPE mean CST >20 µm), from 65.3 to 66.5 letters in eyes with partial EZ attenuation (i.e., EZ-RPE mean CST >0 and ≤20 µm), and from 55.8 to 59.8 letters in eyes with total EZ attenuation (i.e., EZ-RPE mean CST = 0 µm) (Figure 4). At week 48, mean BCVA in eyes without fluid and with maintained EZ integrity (78.4 letters) was significantly greater than in eyes without fluid + partial EZ attenuation (66.5 letters; *p* < 0.0001) and eyes without fluid + total EZ attenuation (55.8 letters; *p* < 0.0001). The mean BCVA in eyes without fluid + partial EZ attenuation (66.5 letters) was significantly greater than in eyes without fluid + total EZ attenuation (55.8 letters; *p* = 0.012), indicating that central subfield EZ integrity in eyes with no fluid was a robust biomarker of visual acuity, as this trend was evident at all time points.

### 3.3. Functional Outcomes Based on Fluid Dynamics and Ellipsoid Zone Integrity

#### 3.3.1. Eyes That Achieved and Maintained No Fluid

Eyes that achieved and maintained no fluid—i.e., those that achieved complete IRF and SRF resolution by week 16 (inclusive) and maintained complete fluid resolution through week 48—had a mean BCVA of 70.8 letters (SD = 15.0; *n* = 171) at week 48. However, among eyes that achieved and maintained no fluid, mean BCVA in eyes with EZ integrity maintained at week 48 was significantly greater than in eyes with EZ integrity total attenuation (78.1 vs. 55.7 letters; *p* < 0.0001) and eyes with EZ integrity partial attenuation (78.1 vs. 66.9 letters; *p* < 0.0001). Moreover, among eyes that achieved and maintained no fluid, mean BCVA in eyes with EZ integrity partial attenuation was also significantly greater than in eyes with EZ total attenuation (66.9 vs. 55.7 letters; *p* = 0.0487; Figure 5).

#### 3.3.2. Eyes with Any SRF or IRF at All Time Points

In eyes with any amount of SRF detected at all time points, mean BCVA at week 48 was 67.5 letters (SD = 16.2; *n* = 33). This was similar to mean BCVA in eyes that achieved and maintained no fluid (70.8 letters; *p* = 0.2816) but was lower than in eyes that achieved and maintained no fluid while concurrently maintaining EZ integrity (78.1 letters). In contrast, mean BCVA at week 48 in eyes with any amount of IRF detected at all time points was 59.0 letters (SD = 17.6; *n* = 30), significantly lower than that of eyes that achieved and maintained no fluid (*p* = 0.0002).

## 4. Discussion

This post hoc analysis of the HAWK phase 3 clinical trial utilized a deep learning-enhanced OCT segmentation platform to investigate the effect of fluid presence across the macula and EZ integrity on visual acuity in patients with nAMD who were treated with anti-VEGFs. Visual acuity was better in eyes without fluid than in eyes with IRF and was similar between eyes without fluid and eyes with SRF. Among eyes without fluid, visual acuity was better in eyes with EZ integrity maintained than in eyes with partial or total EZ attenuation. Importantly, eyes without fluid and with EZ integrity preservation were found to have the best visual acuity overall. This was consistently observed throughout the study and as early as week 4 (Figure 4), suggesting that pre-existing outer retinal atrophy may contribute to lack of visual acuity recovery in certain eyes without fluid. This is a critical insight into the treatment approach and tolerance of fluid in nAMD management and suggests that targeting a dry retina should be of primary importance.

Results from the present analysis are consistent with previous findings demonstrating poorer visual acuity in eyes with IRF present [5,6,11]. The results of the present analysis suggest that SRF does not inherently result in poorer outcomes when compared with the entire group of eyes that achieve fluid resolution. However, it is vital to understand that eyes that achieve fluid resolution and have preservation of central EZ integrity have the best overall BCVA outcomes.

The results of the present study are in line with recent studies demonstrating that EZ integrity correlates with retinal function in nAMD and dry AMD [8,10,12]. In addition, in eyes with macular edema secondary to central retinal vein occlusion or hemi-retinal vein occlusion in the SCORE2 trial, Etheridge and colleagues demonstrated that EZ status (normal, patchy, absent) correlated with the visual acuity score and concluded that EZ assessment can be used to identify patients with poor response to treatment [13]. Taken together, disruption of EZ integrity plays an important role in vision and appears to be a major differentiating factor in visual outcomes for eyes that achieved a dry retina with anti-VEGF therapy for nAMD. The present study also extends previous investigations showing that greater fluctuation in retinal fluid is associated with poorer recovery of EZ integrity in eyes with nAMD during the maintenance phase of anti-VEGF therapy [10,14].

The precise measurement of EZ integrity and retinal fluid compartments is facilitated by advances in OCT imaging, as achieved in the present analysis. By leveraging machine learning-based segmentation algorithms, EZ measurement is no longer a subjective practice; standardized segmentation criteria should make it possible to reproduce classification with high fidelity across studies and clinics, ultimately improving outcomes and disease understanding across a range of disease states [9,10,14,15,16,17].

There are important limitations that should be acknowledged regarding this analysis: it was exploratory, and all significant findings were post hoc and not corrected for multiple comparisons.

## 5. Conclusions

This analysis highlights the potential role of key imaging biomarkers, such as EZ integrity, to better understand outcomes in nAMD. Specifically, EZ integrity is a major factor associated with visual acuity outcomes, and pre-existing EZ attenuation/photoreceptor damage likely limits the extent to which treatment can improve visual outcomes and influences the comparative assessment in outcomes in eyes with and without fluid.

## Figures and Tables

**Figure 1 diagnostics-15-01815-f001:**
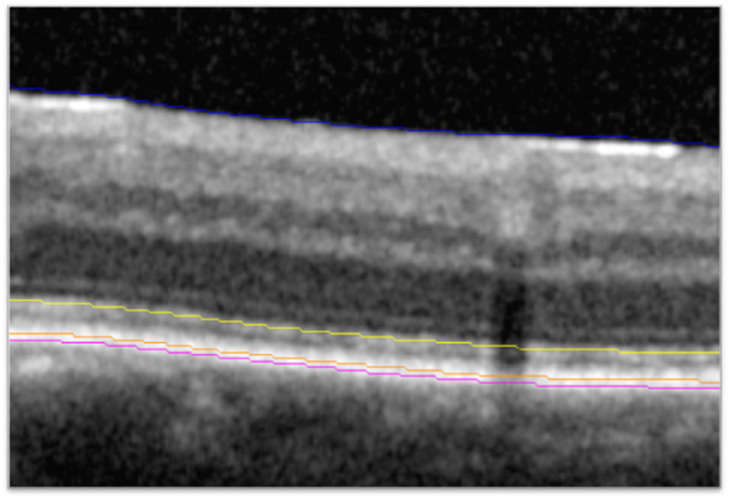
Identification of the EZ and RPE bands. SD-OCT B-scan showing line segmentation of the ILM (blue), EZ (yellow), RPE (orange), and Bruch’s membrane (magenta). EZ, ellipsoid zone; ILM, internal limiting membrane; OCT, optical coherence tomography; RPE, retinal pigment epithelium.

**Figure 2 diagnostics-15-01815-f002:**
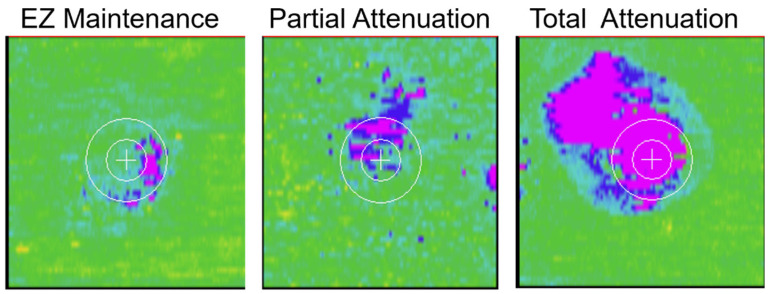
Example of en face EZ-RPE thickness maps. Left: EZ integrity maintained (EZ-RPE CST > 20 μm); middle: partial EZ attenuation (0 μm < EZ-RPE CST ≤20 μm); right: total EZ attenuation (EZ-RPE CST = 0 μm). The inner white circle is the central subfield (1 mm diameter area centered on the fovea). Magenta color shows 0 μm EZ-RPE thickness. CST, central subfield thickness; EZ, ellipsoid zone; RPE, retinal pigment epithelium.

**Figure 3 diagnostics-15-01815-f003:**
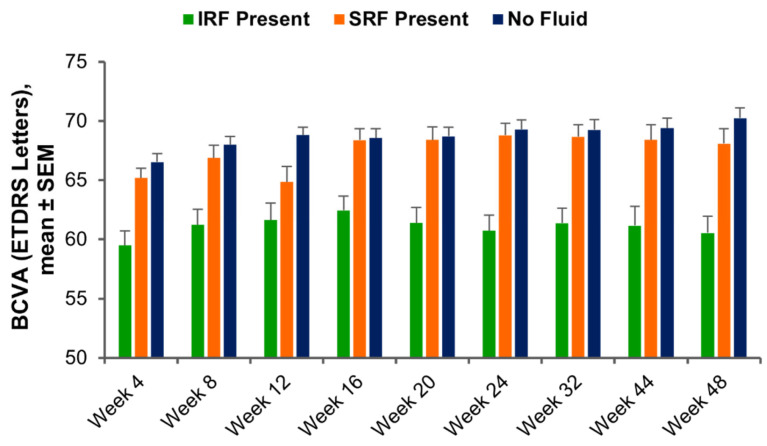
BCVA at each visit after anti-VEGF treatment, in eyes with IRF, SRF, or no fluid at specific time points. BCVA, best-corrected visual acuity; ETDRS, Early Treatment Diabetic Retinopathy Study; IRF, intraretinal fluid; SEM, standard error of the mean; SRF, subretinal fluid; VEGF, vascular endothelial growth factor.

**Figure 4 diagnostics-15-01815-f004:**
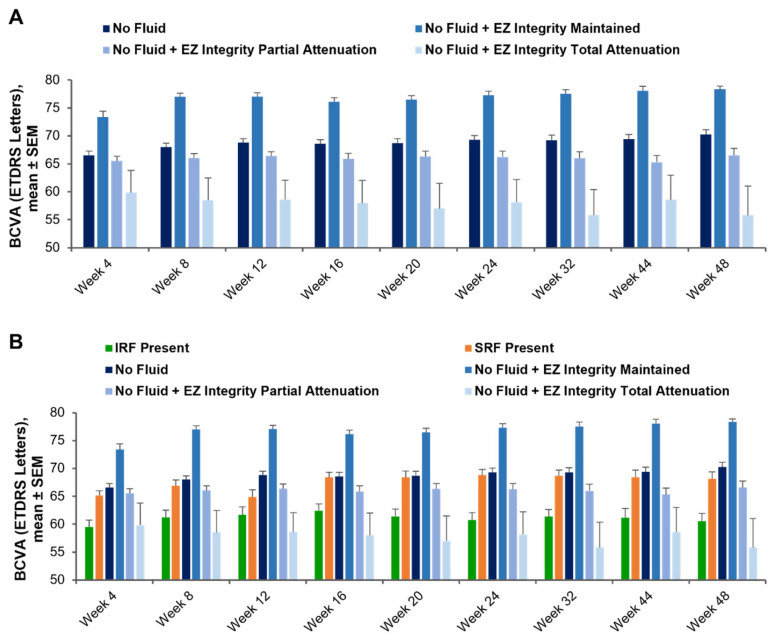
BCVA and EZ integrity. (**A**) BCVA at each visit after anti-VEGF treatment in eyes with no fluid, no fluid + EZ integrity maintained (EZ-RPE mean CST >20 μm), no fluid + EZ partial attenuation (EZ-RPE mean CST >0 and ≤20 μm), or no fluid + EZ total attenuation (EZ-RPE mean CST = 0 μm). (**B**) BCVA at each visit after anti-VEGF treatment, in eyes with IRF, SRF, no fluid, no fluid + EZ integrity maintained, no fluid + EZ partial attenuation, or no fluid + EZ total attenuation. BCVA, best-corrected visual acuity; CST, central subfield thickness; ETDRS, Early Treatment Diabetic Retinopathy Study; EZ, ellipsoid zone; EZ-RPE, ellipsoid zone retinal pigment epithelium; IRF, intraretinal fluid; SEM, standard error of the mean; SRF, subretinal fluid; VEGF, vascular endothelial growth factor.

**Figure 5 diagnostics-15-01815-f005:**
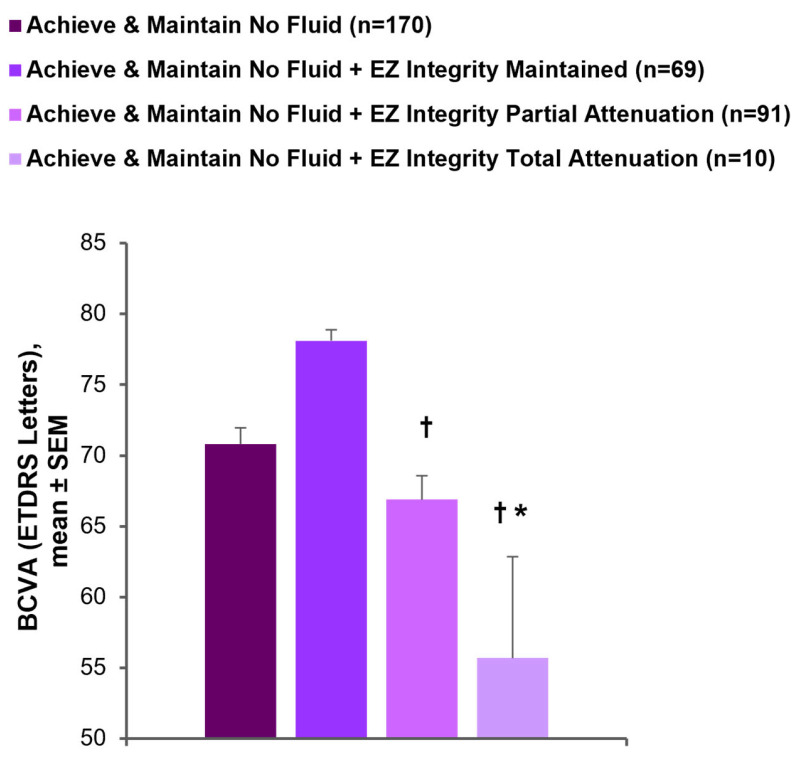
BCVA and EZ integrity at week 48. BCVA at week 48 in eyes that achieved and maintained no fluid (weeks 16–48), no fluid + EZ integrity maintained, no fluid + EZ partial attenuation, and no fluid + EZ total attenuation. * *p* < 0.05 vs. + EZ integrity partial attenuation. † *p* < 0.001 vs. + EZ integrity maintained. BCVA, best-corrected visual acuity; ETDRS, Early Treatment Diabetic Retinopathy Study; EZ, ellipsoid zone; SEM, standard error of the mean.

## Data Availability

The original contributions presented in this study are included in the article. Further inquiries can be directed to the corresponding author.

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
