# Peer review of "Differential Visual Outcomes in Neovascular AMD Based on Ellipsoid Zone Integrity and Fluid Presence: Insights from a Phase III Trial"

_diagnostics, 2025, doi:10.3390/diagnostics15141815_

Round 1
Reviewer 1 Report
Comments and Suggestions for Authors
This is a post hoc treatment analysis of the phase 3 HAWK trial.
This is a well written manuscript and suggests the the EZ integrity is an important biomarker for BCVA following antiVEGF treatment.
The methods could be further clarified:
- The presence of blood and extensive IRC may make OCT grading difficult. Over the study what proportion of OCT scans did not have sufficient image quality to be adequately graded with respect to EZ integrity?
- With respect to SRF/ IRC determination and subsequent characterisation, was the whole of the macular scan considered or was it just the central subfield?
With respect to the analysis, in eyes that had residual IRC or SRF at each time point, was there a correlation with the integrity of EZ and the BCVA or was this only visible in eyes that were dry?
Author Response
Reviewer 1 Comments and Suggestions for Authors
This is a post hoc treatment analysis of the phase 3 HAWK trial.
This is a well written manuscript and suggests the the EZ integrity is an important biomarker for BCVA following antiVEGF treatment.
The methods could be further clarified:
1. The presence of blood and extensive IRC may make OCT grading difficult. Over the study what proportion of OCT scans did not have sufficient image quality to be adequately graded with respect to EZ integrity?
We thank the reviewer for pointing out the need for clarification. In the HAWK trial, n=360 eyes were included in the brolucizumab 6 mg arm and n=360 eyes were included in the aflibercept 2 mg arm. The present analysis excluded 30 eyes from the brolucizumab 6 mg arm and 38 eye in the aflibercept 2 mg arm due to various reasons including image quality, availability, and compatibility. This has been clarified in the manuscript on lines 93-96 of the Methods.
2. With respect to SRF/ IRC determination and subsequent characterisation, was the whole of the macular scan considered or was it just the central subfield?
The entire macular scan was considered in the analysis of retinal fluid. This has been clarified in the manuscript on lines 151-152 of the Results and line 215-216 of the Discussion. This is also described in the Methods on lines 113-114.
With respect to the analysis, in eyes that had residual IRC or SRF at each time point, was there a correlation with the integrity of EZ and the BCVA or was this only visible in eyes that were dry?
The analysis of EZ integrity in eyes with retinal fluid was not conducted. We chose to focus on EZ integrity in eyes without fluid because current management approaches for neovascular AMD aim to evaluate and reduce retinal fluid to preserve visual acuity. In our study, we demonstrate that EZ integrity is an important variable that affects visual acuity even when zero fluid has been achieved.
Previous studies have shown that greater fluctuations in retinal fluid are associated with poorer EZ integrity outcomes (Ehlers JP, et al. Invest Ophthalmol Vis Sci. 2022;63(6):17; Ehlers JP, et al. Ophthalmol Retina. 2024;8(8):765-777). We have added this to the Discussion on lines 240-242. For this reason, it may be the case that fewer eyes with retinal fluid have EZ integrity maintained.
Reviewer 2 Report
Comments and Suggestions for Authors
Dr. Ehlers and colleagues present a compelling post hoc analysis examining the relationship between ellipsoid zone (EZ) integrity, retinal fluid status, and visual acuity outcomes in eyes with neovascular AMD. The results add important insights to the ongoing discussion of fluid tolerance, particularly the differential impact of SRF versus IRF, and highlight the prognostic value of EZ integrity. I have a few comments below:
1.
Since IRF and SRF can be variably distributed (e.g., foveal vs. extrafoveal), do the authors have any data on whether the location of the fluid influences the relationship between EZ integrity and BCVA?
2.
Given the dynamic nature of the fluid response and the potential for structural recovery, it would be valuable to understand whether EZ integrity improves or deteriorates over time based on fluid resolution or persistence. Do the authors have longitudinal data on the evolution of EZ status in response to changes in fluid compartments?
Author Response
Reviewer 2 Comments and Suggestions for Authors
Dr. Ehlers and colleagues present a compelling post hoc analysis examining the relationship between ellipsoid zone (EZ) integrity, retinal fluid status, and visual acuity outcomes in eyes with neovascular AMD. The results add important insights to the ongoing discussion of fluid tolerance, particularly the differential impact of SRF versus IRF, and highlight the prognostic value of EZ integrity. I have a few comments below:
- Since IRF and SRF can be variably distributed (e.g., foveal vs. extrafoveal), do the authors have any data on whether the location of the fluid influences the relationship between EZ integrity and BCVA?
We thank the reviewer for the suggestion. We unfortunately do not have an analysis that examines the impact of retinal fluid location on the relationship between EZ and BCVA.
- Given the dynamic nature of the fluid response and the potential for structural recovery, it would be valuable to understand whether EZ integrity improves or deteriorates over time based on fluid resolution or persistence. Do the authors have longitudinal data on the evolution of EZ status in response to changes in fluid compartments?
We unfortunately do not have this analysis for the present dataset. However, in previous publications our group showed less improvement in EZ integrity with anti-VEGF treatment in eyes with greater fluctuations of retinal fluid (Ehlers JP, et al. Invest Ophthalmol Vis Sci. 2022;63(6):17; Ehlers JP, et al. Ophthalmol Retina. 2024;8(8):765-777). We have added this to the Discussion on lines 240-242.